# Survival Rates of Glass versus Hybrid Ceramics in Partial Prosthetic Restorations: A Scoping Review with Emphasis on Adhesive Protocols

**DOI:** 10.3390/jcm12216744

**Published:** 2023-10-25

**Authors:** Manuela Manziuc, Alex Abbas Khechen, Marius Negucioiu, Irina Poiană, Andreea Kui, Anca Mesaroș, Smaranda Buduru

**Affiliations:** 1Prosthetic Dentistry and Dental Materials Department, Iuliu Hatieganu University of Medicine and Pharmacy, 32 Clinicilor Street, 400006 Cluj-Napoca, Romania; manuelamanziuc@yahoo.com (M.M.); smarandabudurudana@gmail.com (S.B.); 2Faculty of Dental Medicine, Iuliu Hatieganu University of Medicine and Pharmacy, 8 Victor Babeș Street, 400008 Cluj-Napoca, Romania; 3Cluj County Emergency Clinical Hospital, 3-5 Clinicilor Street, 400006 Cluj-Napoca, Romania

**Keywords:** partial prosthetic restorations, CAD/CAM techniques, glass–ceramic systems, hybrid ceramics, adhesive techniques

## Abstract

As dental practices and methodologies evolve, the emergence of novel materials adds complexity to clinical choices. While glass ceramics, particularly those based on lithium disilicate and leucite-reinforced variants, have been extensively researched and are well regarded for their attributes, hybrid ceramics remain relatively recent area of research that is less investigated. This review aims to evaluate the durability of glass and hybrid ceramics while assessing the role of various adhesive techniques on restoration longevity. Using a comprehensive search of PubMed and EMBASE, 84 articles from the past decade were found. Only eleven met the set criteria for analysis. The results underscore the urgent need for the extended monitoring of partial prosthetic restorations. The existing literature has significant gaps, hindering the attainment of dependable insights about these materials’ long-term performance. For a clearer understanding of how different ceramic systems affect restoration survival rates, rigorous research involving more participants and uniform outcome documentation is vital.

## 1. Introduction

Ceramic systems, known for their esthetic appeal and biocompatibility, have a significant role in the field of restorative dentistry. While there are several classification systems for ceramic materials available in the literature [1,2], the one emphasizing microstructure is simpler and more advantageous from a clinical point of view. this classification divides ceramic materials into three large classes [3,4]: glass ceramics and their subclasses, polycrystalline ceramics, and hybrid ceramics. These classes have replaced the now-outdated infiltrated aluminous ceramics.

Glass ceramics have excellent esthetic properties and adequate mechanical resistance. They are composed of a crystalline matrix that is responsible for their strength and is the majority component included in a glassy matrix [3,4]. They can be further classified (Figure 1) into subclasses, such as feldspathic ceramics, leucite-reinforced feldspathic ceramics, lithium silicate ceramics, and lithium disilicate ceramics, that have different mechanical, esthetic, and optical properties, offering more choices for esthetic treatments depending on the location of the tooth, the piece to be produced, and the patient’s esthetic demands [5].

Developed as an innovative material group, hybrid ceramics have not yet been extensively researched concerning their long-term clinical implications. Primarily composed of polymer-infiltrated ceramic–network (PICN) materials, this innovative ceramic system was created to be used exclusively within CAD/CAM digital technology. Microstructurally, it presents an inorganic ceramic matrix reinforced with an organic polymeric material, with a perfect interpenetration between the two chemical structures [6]. Vita Enamic (Vita Zahnfabrik) represents the material of choice for this all-ceramic system, with particular optical and mechanical properties. Due to the polymer matrix, it presents a modulus of elasticity of 30 GPa [6], very close to that of dentin 15–20 GPa [7,8,9] but lower than that of dental ceramics, and a resistance to flexural forces of 150–160 MPa [6], which is inferior to lithium disilicate (342 MPa). Vita Enamic is easy to mill, and restorations can have a lower thickness and greater accuracy of marginal geometry compared to restorations obtained by milling silicate ceramics. This ceramic material is commercialized in two translucencies, HT (increased translucency) and T (translucent), in different chromatic shades that correspond to the Vita 3D-MASTER color selection system. Vita Enamic is indicated for minimally invasive dental restorations, such as veneers, inlays, onlays, partial crowns, anterior and posterior crowns, and both on natural teeth and on dental implants [6].

Nevertheless, nano-ceramics or composite resins reinforced with nanoceramic crystals are also hybrid dental materials with physical and optical properties that combine the characteristics of polymer materials with those of dental ceramics. Microstructurally, they present an organic polymeric matrix reinforced with ceramic inorganic crystals, such as quartz and zirconium dioxide. Ceramic crystals improve the mechanical properties of the material, which has greater resistance to fracture and wear, being indicated for dental restorations in cases of patients with occlusal parafunctions, such as bruxism [8]. These dental materials are indicated for the fabrication of veneers, inlays, onlays, and crowns on natural teeth or implants [6]. Lava Ultimate (3M ESPE), Cerasmart (GC), Shofu Block HC (Shofu), Grandio Blocs (VOCO), and KATANA AVENCIA Block (Kuraray Noritake Dental, Inc) are the representative materials for this category of dental materials [9,10,11].

Considering the optical properties of nano-ceramics, Paolone et al. concluded that while CAD/CAM resin-based blocks show higher color stability compared to direct/indirect resin-based composite, and that they exhibit lower color stability than ceramic materials. Nevertheless, the color stability is also influenced by material composition, staining media, as well as finishing/polishing procedures [12]. In another experimental study, the translucency of CAD/CAM materials (Tetric CAD (TEC) HT/MT, Shofu Block HC (SB) HT/LT, Cerasmart (CS) HT/LT, Brilliant Crios (BC) HT/LT, Grandio Bloc (GB) HT/LT, Lava Ultimate (LU) HT/LT, and Katana Avencia (KAT) LT/OP.) and printable composite materials (Permanent Crown Resin) for fixed dental prostheses was tested. The authors concluded that because there is a significant range of reported translucency values, clinicians should exercise caution when choosing the most appropriate material, especially considering factors such as substrate masking and the necessary clinical thickness [13].

Dental techniques and sciences are constantly evolving, resulting in an increasing number of material choices that can complicate clinical decisions. New questions are being raised following the rise of adhesive techniques [14,15,16,17,18,19,20,21], improvements in the mechanical properties of glass ceramics [22], and the introduction of hybrid ceramics to our profession.

In this context, survival rates of dental restorations performed from newer materials is an important aspect as they will provide an overview of the material’s long-term reliability, clinical efficacy, and potential as a sustainable choice in restorative dentistry. Evaluating these rates not only ensures the safety and satisfaction of patients but also guides dental professionals in making informed decisions about the adoption of innovative materials and techniques in their practice. The modified United States Public Health [23] Services (USPHS) criteria serve as a standardized evaluation system for assessing the clinical performance of dental restorations.

Therefore, the aim of this scoping review was to critically evaluate and compare the durability and longevity of glass ceramics and hybrid ceramics, emphasizing their recent technological advancements. In addition, another scope of this review was to assess the impact and effectiveness of different adhesive technologies on the survival rate and overall performance of both types of material systems in dental applications.

## 2. Materials and Methods

This study represents a scoping review. The research protocol was formulated following the guidelines provided by the PRISMA Extension for Scoping Reviews (PRISMA-ScR) [24].

### 2.1. Eligibility Criteria

The inclusion criteria were as follows: (1) clinical trials involving patients who have received partial restorations, glass ceramics, or hybrid ceramics; (2) clinical evaluation of survival must be performed, analyzing predefined criteria by the author; (3) in vitro studies or lab experiments involving different types of partial prosthetic restorations; (4) articles written in English; and (5) publications published between 2013 and 2023.

The following exclusion criteria were considered: (1) other types of reviews or systematic reviews; and (2) non-accessible publications.

### 2.2. Information Sources

A structured electronic search was conducted between June 2023 and July 2023, over the last 10 years, in the following databases: PubMed, and Embase. MeSH and Emtree terms were used, where applicable. In addition, a handsearching of relevant studies was performed. All references were imported and organized in the Mendeley online software.

### 2.3. Search Strategy

Articles selection was conducted in three phases. To include all possible retrievable studies, the search was performed using three different combinations of terms, such as “Inlay” OR “Onlay” OR “Overlay” OR “Veneers” AND “Glass ceramics” OR “Polymer-infiltrated ceramic” OR “Hybrid ceramic” AND “Survival”. Articles published in the last 10 years (2013–2023) were considered. After retrieving all articles, three databases were created using Mendeley software that allowed for the organization of the publications and perform an independent blind screening of the included studies. Three researchers independently performed the search and scored the ratings. When in doubt about including a specific study, the researchers discussed between them and a fourth one was asked for debate. All references were managed using Mendeley Reference Manager v2.101.0.

### 2.4. Scoring Systems Used for Paper Evaluation

In order to include relevant articles based on this search, we have developed a scoring system based on five categories and their corresponding sub-criteria. Each category of the evaluated studies would be assessed with a potential score (Table 1).

## 3. Results

### 3.1. Data Collection

A total of 84 articles were enrolled after applying the search strategy (Table 2). After the elimination of duplicates and eliminating the ones not related to the topic, 25 records were considered for screening. During the first phase, the included articles were selected based on their titles/abstracts and their relation to the study question. Therefore, the screening process generated 25 articles, and 24 publications were further assessed for eligibility. Any disagreements were resolved by discussion and by consultation with a fourth one. Finally, a total of eleven publications were included in this review.

The selection process, along with the inclusion decision, is shown in Figure 2, the PRISMA flow diagram.

The main characteristics of the studies that were considered in this review are summarized in Table 3.

### 3.2. Description of the Studies and Analysis

From the total of eleven articles included in this scoping review, three articles investigated veneers, one studied inlays and overlays, three studied inlays and onlays, one concerned overlay restorations, two concerned onlays, and one considered inlays and overlays. Several materials were compared, including hybrid ceramics, lithium disilicate-based ceramics, leucite-reinforced glass ceramics, lithium silicate reinforced with zirconia, and feldspathic ceramics. The follow-up period for the restorations varied, ranging from 2 years for the shortest period to 20 years for the longest. The methods of implementing the materials may differ between studies, with some using blocks manufactured by CAD-CAM technology, while others implement prosthetic pieces using pressed ceramics. The adhesion of the ceramic was performed either by a bonding technique or by self-etching. It is interesting to compare if there is a difference in survival between these two techniques.

The majority of studies included in this review used the “modified United States Public Health Services criteria” (USPHS), except for Al-Akhali [20], who studied the influence of thermomechanical fatigue on the fracture resistance of occlusal facets, and Malament [30], who investigated the survival of 10.9 years of partial lithium disilicate restorations, depending on different factors. Originally developed by the USPHS, these criteria have undergone modifications over the years to better suit the evolving nature of dental materials and techniques. The criteria encompass various parameters, including retention, color match, marginal adaptation, anatomic form, surface texture, and postoperative sensitivity, among others. Each parameter is graded based on its clinical presentation, allowing for a comprehensive assessment of the restoration’s functionality, esthetics, and longevity. By employing the modified USPHS criteria (Table 4), clinicians and researchers can objectively compare the performance of different dental materials and techniques, ensuring that patients receive restorations of the highest quality [23].

Some studies used a numerical rating system from 0 to 5, similar to USPHS. There are minor differences in the evaluation system used by different studies, with some evaluating more criteria (such as secondary caries, marginal adaptation, joint integrity, surface condition, color, and anatomical shape, depending on their clinical performance). Some studies also evaluate postoperative sensitivity, restoration, and tooth fractures as well as patient satisfaction. The follow-up process differed between the studies included, as presented in Table 5.

The definitions of survival or failure of a restoration differ from one study to another (Table 6), depending on the authors. Spitznagel et al. [25] define the absolute failure of a restoration as a “clinically unacceptable fracture” that leads to the replacement of the piece. A Charlie rating for joint integrity, marginal adaptation, marginal caries, or detachment of the ceramic piece is also considered a failure. They also define cases as a “relative failure” in which there are minor cohesive fractures, slight material cracks, slight joint discoloration, or minor marginal adaptation defects. For Aslan et al. [26], the presence of caries, the detachment of the veneer, and fractures are considered failures.

In the study written by Al-Akhali et al. [28], restorations were subjected to in vitro thermomechanical cycles. Success was characterized by the absence of macroscopic deterioration. They also defined partial success as the presence of some cracks but that do not affect the integrity or adhesion of the prosthetic piece. In cases where there is fracture and/or detachment, it is considered a failure. Lu et al. [29] did not clearly define failure, but those found were caused by ceramic fracture or detachment. Taschner et al. [30] characterized restorations that had to be replaced, whether due to the piece itself (ceramic fracture) or to the dental substrate (enamel fracture), and were considered failures.

Malament et al. [31] defined failure as ceramic fracture that led to replacement. They also noted that in some cases, the restoration had to be replaced not due to failure but because of the loss of an adjacent tooth, which transformed the restored tooth into a pillar for fixed prosthesis. In Rinke et al. [34], success meant that the restoration did not require intervention, and survival meant that the restoration underwent intervention but did not present an absolute failure. Absolute failure was defined as tooth fracture, caries, periodontal disease, or complete ceramic fracture, all of which led to restoration replacement. For Guess et al. [33], absolute failures were unacceptable ceramic fractures, secondary caries, and endodontic complications. Relative failures were acceptable minimal cohesive fractures, detachment, and the presence of a Charlie rating in the USPHS criteria. Xiao et al. [32] did not clearly define failure, but the those encountered concerned ceramic fractures that automatically led to a Charlie rating in the USPHS criteria. Ideally, studies should have had the same definition for a more precise quantification of survival based on materials. Table 4 shows that regardless of the type of restoration, materials used, or adhesive protocol followed, the vast majority of failures are due to bulk fracture of the material or the detachment of the ceramic piece. Of the 59 absolute failures, 37 (62.71%) were due to ceramic fracture, and 12 were due to detachment (20.34%).

### 3.3. Adhesion Protocol

Regarding adhesion, Peumans [35] et al. compared the performance of the self-adhesive resin RelyX Unicem with and without etching, revealing that the addition of etching and rinsing to the adhesive protocol slightly improved the marginal adaptation over a 4-year follow-up. Similarly, Taschner et al. [30] compared two adhesive protocols and found variations in marginal adaptation over time, emphasizing the influence of the adhesive joint on color concordance between the restoration and the tooth.

## 4. Discussion

This scoping review aimed to investigate the survival rate of glass ceramic and hybrid ceramic restorations, along with evaluating the impact of adhesion protocols in the restorations durability. Most of the studies included in our research used the USPHS criteria to evaluate the restorations during follow-up. These aspects allowed for a comparison between the results obtained, the success rate, as well as the clinical performance of esthetic restorations.

### 4.1. Marginal Adaptation and Joint Integrity

The results obtained by Spitznagel et al. [25] show that at 36 months, 86.4% of PCR and 88.6% of PICN inlays were rated Alpha. 13.6% of PCR and 11.4% of inlays received a Bravo rating. It can be seen that no Charlie rating was recorded for this criterion during the follow-up period. Coşkun et al. [22] did not find a statistically significant difference between the two materials used (lithium disilicate and hybrid ceramic). For the E.max group, all restorations were still rated Alpha after 2 years, while for the hybrid ceramic group, only one restoration had moved to Bravo for this criterion. Lu et al. [28] found that at 0.5 years, 4.5% of hybrids (three restorations) and 5.9% of feldspathic ceramics (two restorations) were rated 1. At 3 years, 7.7% (five restorations) of hybrids and 6.9% (two restorations) of feldspathic ceramics were rated 1. These three authors included hybrid ceramics in their studies, which allows for a first judgment regarding the difference between hybrid ceramics and glass ceramics.

Coşkun et al. [27] showed better results than Spitznagel [25] and Lu [28], but this could be due to the fact that the follow-up period is 1 year shorter than the latter and possibly also because the former compared a hybrid ceramic and lithium disilicate, which is recognized by clinicians for its qualities, compared to the feldspathic ceramic used by Lu [28], which is not at the same level in terms of mechanical qualities. However, the absence of a statistically significant difference between the groups studied by Coşkun et al. [27] is a notable success.

In the article by Xiao et al. [33], lithium disilicate onlays received two Bravo ratings at 6 months, three Bravo at 12 months and one Bravo at 24 months. It was compared two types of preparations for onlays in lithium disilicate, one with a marginal shoulder (group S) and the other with a bevel encircling the preparation (group B). The integrity and marginal adaptation were superior for group B in the molar region. This is attributed by the author to the increase in occlusal stress in group S due to the shoulder, which, along with the forces induced on the restoration, causes fractures in the ceramic along the marginal joint. These stresses are not as important in beveled preparations. The affected restorations were repaired and followed up, and they did not show problems thereafter.

Peumans et al. [35] compared the performance of the self-adhesive resin RelyX Unicem when used normally, without etching (group NE), and in a classic protocol with etching and rinsing (group E). The marginal adaptation after 4 years of follow-up decreased significantly for both groups without a statistically significant difference between the two. The ratings corresponded to Alpha 1 in 6.7% of the restorations in group E and 3.3% in group NE. The majority were Alpha 2, 13.3% in group E, and 23.3% in NE were Bravo (these two notes showing clinically acceptable adaptation). The restorations noted as Charlie (one), and two others were Delta, referring to three failures. This criterion underwent the greatest decrease in quality over the 4-year follow-up. Despite the absence of a statistically significant difference between the two groups for this criterion, the author indicates that the performance of group E is still slightly above group NE, which is visible by a decreased percentage of Bravo rating for group E compared to group NE.

Taschner et al. [30] compared two adhesive protocols, one using a self-adhesive resin, RelyX Unicem (group RX), and the other using Variolink II-low (group SV), and found no significant difference in marginal adaptation after 14 years. However, at R4 (2 years of follow-up), group SV performed better than group RX for this criterion, and it was only at this point in the follow-up that a significant difference existed between the two groups. The absence of Alpha 1 is mostly due to an excess of bonding resin at the marginal level. However, group RX undergoes a greater loss of resin at the bonding joint compared to group SV. At 14 years, 50% of SV and 39% of RX are in Alpha 2, 50% of SV and 54% of RX are in Bravo, and 0% of SV and 7% of RX are in Charlie. Even though there is no statistically significant difference, group SV shows better results.

Aslan et al. [26] found that 12.8% of restorations (veneers) received a score of 1 (equivalent to Bravo) for marginal adaptation, with recessions being the cause. In total, 3.63% of restored teeth showed a 1 mm gingival recession, and 1.1% showed a 2 mm gingival recession. These recessions are, according to the author and Dumfahrt and Schaffer [36], a result of age, as these studies were conducted over a long-time interval. Guess et al. [32], who compared two types of preparations for veneers, one with overlap (group OV) and the other with full veneer (group FV), saw a marked decrease in the quality of marginal adaptation over 7 years without a statistically significant difference between the two types of preparation. The values of the two groups, which were 100% Alpha at baseline, decreased to 28% for group OV and 20% for group FV, with the rest of the restorations being rated Bravo.

For the integrity of the joint, Spitznagel et al. [25] noted that 86.4% of PCR and 91.4% of inlays remained in Alpha. In total, 13.6% and 8.6% were, respectively noted as Bravo. No Charlie notation was recorded during the follow-up period. The author affirms that the decrease in joint integrity as well as marginal adaptation plays a role and contributes to the failure of restorations. For joint integrity, at 2 years, the results of Coşkun et al. [27] are the same for the two groups of materials (lithium disilicate and hybrid ceramic). For each material, two restorations were noted Bravo, representing 6.7% of each group. These results are close to the previous study by Spitznagel [25], which once again gives hybrid ceramics a chance to establish themselves as durable and reliable materials.

According to Lu et al. [28] at 0.5 years, 10.4% of PICN (seven restorations) and 14.7% of Mark II were noted as 1. two PICN restorations (3.0%) were noted as 2. At 3 years, 10.8% of PICN and 17.2% of Mark II were noted as 1. In total, 4.6% of PICN (three restorations) were noted as 2, and no Vita Mark II received this notation for this criterion. Xiao et al. [33] did not find a difference between groups S and B. The color of the marginal joint was affected but remained acceptable in five restorations that received a Bravo notation at 24 months, two in group S, and three in group B.

For Aslan et al. [26], 11.6% of the veneers showed a slight discoloration of the joint, a sign of a resin deficit and joint infiltration. Guess et al. [32] explained the discoloration and loss of integrity of the resinous joint and, among other things, marginal adaptation by the aging of the bonding resin, which marginally deteriorates and becomes prone to discoloration/infiltration. In their case, the values fall after 7 years to 22% Alpha for group OV and 20% for group FV, with the rest of the restorations being classified as Bravo, which constitutes a clear decrease for the groups, regardless of the type of preparation studied.

### 4.2. Surface Integrity

Surface integrity is an important criterion for evaluating the quality of ceramic restorations. Spitznagel et al. [25] observed a decrease in surface state over time, with only 17.5% of PICN restorations being rated Alpha after 6 months, 10.0% after 12 months, and 3.6% after 24 months. This trend continued until the end of the follow-up period, with 97.5% of restorations rated Bravo and 2.5% Alpha. No Charlie rating was recorded over the duration of follow-up. The author states that the most affected locations were the occlusal fissures as well as the contact points.

Coşkun et al. [27] did not find a statistically significant difference in surface state between lithium disilicate and hybrid ceramic inlays/onlays over a 2-year follow-up period. For this criterion and over the entire follow-up period, all restorations in both groups (DL and CH) received an Alpha rating.

Taschner et al. [30] claimed that the weaker performance of RX, which leads to marginal microfractures, is responsible for a significant difference between the groups for the surface state criterion. At R6 (4 years) and R7 (14 years), the surface state of SV group restorations is significantly better than that of RX group restorations. After 14 years, 11% of SV and 3% of RX are Alpha 1, 81% of SV and 68% of RX are Alpha 2, and 8% of SV and 29% of RX are Bravo. These results show a clear difference between the two groups for this criterion, which is therefore affected by the adhesive protocol applied to the tooth substrate.

Aslan et al. [26] noted that the surface state of the ceramic was defect-free for 98.2% of the veneers and therefore received a score of 0. Since veneers are not placed directly in areas of high occlusal forces, such as inlays or onlays [37,38], it is not surprising that the surface quality is higher compared to posterior restorations.

### 4.3. The Color

Regarding the color match criterion, Spitznagel et al. [25] found that the color match between the ceramic and the tooth received Alpha ratings in 97.7% of the PCR and 97.1% of the inlays. Only one restoration in each group received a lower Bravo rating.

Coşkun et al. [27] did not find a significant difference between the two groups (DL and CH) for the color criterion. The results were the same for both groups throughout the follow-up period, with 28 restorations in Alpha and 2 restorations in Bravo.

For onlays evaluated by Lu et al. [28], at 0.5 years, 13.4% (nine onlays) of the PICN and 20.6% of the Mark II (seven onlays) were rated 1. At 3 years, 10.7% of the PICN were rated 1, and 4.6% (three onlays) were rated 2. 24.1% (seven onlays) of the Mark II were rated 1, and only one (3.4%) was rated 2. These results show weaker performance for the Mark II material compared to the other two studies that included similar materials and restoration types. The percentage of Mark II restorations that received a lower rating is relatively higher than for PICN restorations, even though the number of PICN restorations was 55 at the end of the follow-up period (3 years) and 21 for Mark II.

Taschner et al. [26] found that at 4 years, the color and its concordance with the natural tooth color were rated Alpha 1 for 94% of the SV group and 82% of the RX group. These results were significantly reduced at 14 years, with only 35% of the SV and 11% of the RX in Alpha 1. A statistically significant difference was found between the two groups. This difference was due to and accentuated by the adhesive joint for the RX group, which, once destroyed, leaves a space prone to discoloration. Another reason for the decrease in color concordance between the restoration and the tooth is the natural tendency of teeth to darken over time. The adhesive protocol could therefore have an impact on the color criterion and the color concordance between the substrate and the restoration, with the standard adhesive protocol achieving more stable colorimetric stability over time.

The results of Peumans et al. [35] show that for the color criterion, 66.7% of the restorations in the E group and 40% of the restorations in the NE group were still in Alpha 1 after 4 years, demonstrating once again the superiority of adding etching and rinsing to the adhesive protocol. Aslan et al. [26] did not observe any color changes in the veneers during the follow-up period, so all restorations received a score of 0 for this criterion. Guess et al. [32] noted that after 7 years, the color criterion was Alpha in 78% of the OV restorations and 100% in the FV group. However, it is important to note that this result of 100% was probably obtained due to the loss of follow-up of certain restorations since at 70 months (the previous check-up), the results were 95% Alpha for the OV group and 88% for the FV group.

### 4.4. Morphology

Spitznagel et al. [25] noted Alpha for the morphology of PICN restorations throughout the follow-up period for all restorations.

Lu et al. [28] followed the morphology of hybrid and feldspathic ceramic restorations and at 0.5 years, 100% of hybrids received a score of 0, and only one feldspathic restoration (2.9%) received a lower score of 1. At 3 years, 10.8% of hybrids and 10.3% of feldspathic restorations were given a score of 1. The two materials seem to show similar results for this criterion.

The results for lithium disilicate restorations followed by Xiao et al. [33] show that at 6 months, 117 were in Alpha, 1 was in Bravo, and 2 were in Charlie. The Charlie score was attributed to two restorations on the second mandibular molar due to fracture. At 12 months, one was in Bravo and three were in Charlie for the same reason (two maxillary first molars and one second mandibular molar).

Guess et al. [32] show that after 82 months, 80% of FV restorations were rated Alpha for this criterion, and 67% of OV group restorations were rated Alpha as well.

### 4.5. Sensitivity and Patient Satisfaction

Some authors have included perceived dental sensitivity and patient satisfaction.

Coşkun et al. [27] found no significant difference between the two groups (DL vs. CH) regarding patients’ masticatory efficiency. Masticatory efficiency and patient satisfaction were even increased in both groups over the 2-year follow-up period.

Taschner et al. [30] reported 100% satisfaction without complaints after cold tests at 4 years. According to the author, this absence of complaints after 4 years is due to the adhesion of the bonding resin to the dentin, which thus blocks the dentinal tubules and the perception of sensitivity. After 14 years, two patients in each group had “slight complaints”, two patients in the SV group complained about esthetics and chewing comfort. At R7 (14 years), 27% of SV group patients and 29% of RX group patients saw their sensitivity to cold test drop from Alpha 1 to Bravo.

Peumans et al. [35] report that sensitivity, if present at the beginning of the follow-up, disappeared later. Only one patient (NE group) still had sensitivity to chewing and flossing after 4 years, but these sensitivities had decreased.

Aslan et al. [26] reported that 97.1% were satisfied with the functional aspect, and 98% said they were satisfied with the esthetics. A total of 45 restored teeth were sensitive post-bonding. The symptoms disappeared after 3 weeks. In the case of 10 restored teeth, slight sensitivity was still perceptible after 12 months.

All articles show a similar trend towards patient satisfaction, with an improvement in symptomatology over time in cases where it is present.

### 4.6. Fracture Resistance

Al-Akhali et al. [27] conducted a study on the influence of thermomechanical fatigue on the fracture resistance of four CAD-CAM materials used for occlusal veneers: lithium disilicate (DL), zirconia-reinforced lithium silicate (LS), polymer-infiltrated ceramic (PICN), and polymethylmethacrylate (PM). Thin occlusal veneers were fabricated and bonded to extracted intact premolars using a self-etching bonding technique. The restorations had a thickness of 0.5 mm at the fissures and 0.8 mm at the cusps. Half of the restorations in each group were randomly designated to undergo thermomechanical fatigue loading of 1,200,000 cycles with a force of 98 Newton and a temperature variation of 5 to 55 degrees Celsius with a total of 5500 thermal cycles. Fractured or debonded restorations were considered failures, those displaying intrinsic fractures were considered partial successes, and those unaffected were considered successes.

The complete success rate was 50% in the DL group, 25% in the LS group, 0% in the PICN group, and 50% in the PM group.

The partial success rate was 0% in the DL group, 37.5% in the LS group, 37.5% in the PICN group, and 0% in the PM group.

The cumulative survival rate was 50% for the DL group (there were only complete successes in this group), 62.5% for the LS group, 37.5% for the PICN group (there were only partial successes in this group), and 50% for the PM group (there were only complete successes in this group).

No statistically significant difference was found in the cumulative survival rate between the DL, LS, and PM groups. The survival rate of the PICN group was reduced. The lowest force applied at which a failure occurred was 98 N for a PICN restoration, and the highest was 1250 N for an LS group restoration. The author observed that thermomechanical fatigue significantly decreased the fracture resistance of PICN and PM restorations, unlike DL restorations, which were not significantly affected. The author emphasizes that these results were obtained on a substrate treated with a self-etching primer, which reduces the overall survival rate. Although the survival rate seems to be lower for the PICN group, the restorations that failed were able to survive more masticatory cycles than the DL and LS groups due to their increased fatigue resistance.

Lu et al. [28] believe that the easier machining of hybrid ceramic blocks (thus reducing the risk of defects inside the restoration) could give better long-term results in the treatment of posterior teeth, despite the fact that no statistically significant difference was found regarding fractures or debonding between the PICN and feldspathic ceramic groups.

### 4.7. Adhesion

Based on the articles included in our research, adhesion protocols played an important role in the success and longevity of partial prosthetic restorations. The studies highlight the importance of marginal adaptation and joint integrity, both of which are directly influenced by the chosen adhesion method. For example, Peumans et al. [35] evaluated the efficacy of the self-adhesive resin RelyX Unicem, both with and without etching. Their findings indicated that incorporating etching and rinsing into the adhesive process marginally enhanced the marginal fit over a span of 4 years.

In a similar study, Taschner et al. [30] compared two adhesive protocols on the survival rate and obtained variations in marginal fit over time, emphasizing the influence of the adhesive joint on color concordance between the restoration and the tooth. Our research also suggests that the adhesive protocol can impact sensitivity post-restoration, with some studies reporting a decrease in sensitivity over time. Overall, the choice of adhesion protocol is crucial in determining the quality, durability, and patient satisfaction of ceramic restorations.

### 4.8. Clinical Recommendations

Based on the findings of the present scoping review, both lithium disilicate and hybrid ceramics show promising results regarding marginal adaptation and joint integrity. Given the lack of significant difference in performance between these materials in several studies, clinicians can choose based on individual case requirements, considering factors like esthetic demands, occlusal forces, and financial considerations.

Considering the deterioration of surface quality over time, especially in occlusal fissures and contact points, regular monitoring and possibly the application of surface sealants or other protective measures could help maintain the integrity of the restoration’s surface.

In order to ensure color stability over time, particularly for restorations in esthetic areas, an effective adhesive protocol should be used. Since teeth naturally darken over time, the choice of bonding agents that can withstand color changes or materials that maintain color stability is crucial. Regular follow-ups to check color concordance over the years are advisable.

Clinicians should stay updated with the latest research findings, as newer materials and techniques continuously emerge. Incorporating evidence-based practices into clinical protocols is key to optimizing patient outcomes.

## 5. Conclusions

Glass ceramics, especially lithium disilicate based ceramics and leucite-reinforced glass ceramics, are widely studied, and recognized for their qualities.In addition, hybrid ceramics are relatively newer, and there is a crucial lack of data in the literature, making it difficult to obtain reliable and significant results representative for the long-term behavior of these materials.Clinically, the choice between these two families of materials should be made based on different clinical data, as each offers advantages and disadvantages. The material used is one of the factors influencing survival, but it is probably not the most important.Based on initial findings from short-term studies, researchers encourage the use of hybrid ceramic, suggesting that they could be suitable for long-term, durable restorations. As more research emerges in the upcoming years, we anticipate a clearer understanding of this topic.

## Figures and Tables

**Figure 1 jcm-12-06744-f001:**
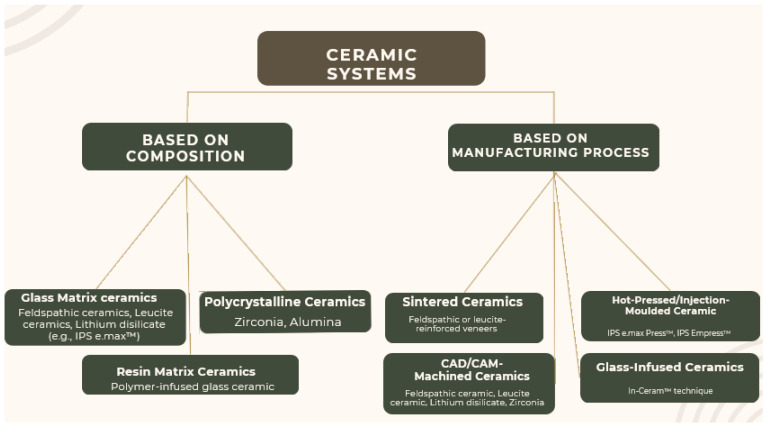
Classification of ceramic systems, according to R. Wassell [5].

**Figure 2 jcm-12-06744-f002:**
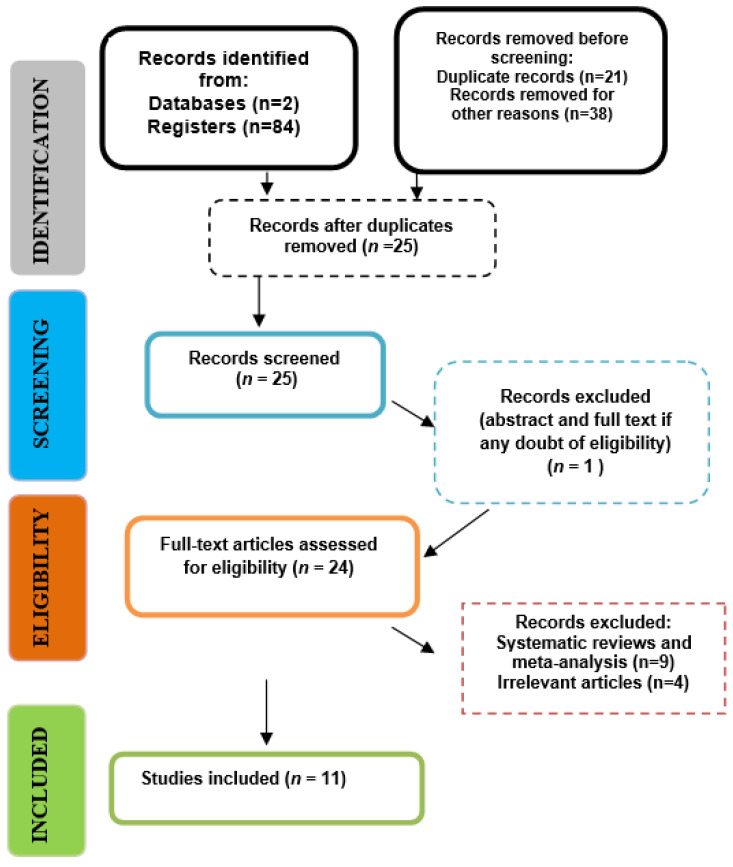
PRISMA flow-chart of the selection process.

**Table 1 jcm-12-06744-t001:** The scoring system used to evaluate the articles included in the research.

Criteria	Sub-Criteria	Score (Points)
Research design	Randomized control trial	25
	Cohort study	20
	Case-control study	15
	Cross-sectional study	10
	Case reports/case series	5
Sample size	>300 subjects	20
	200–299 subjects	16
	100–199 subjects	12
	50–99 subjects	8
	<50 subjects	4
Follow up duration	>10 years	15
	5–10 years	12
	2–5 years	9
	1–2 years	6
	<1 year	3
Outcome reporting	Comprehensive reporting (various outcomes)	20
	Succes and failure rates	10–15
Evaluation criteria	Multiple standardized evaluation methods	20
	One standardized evaluation method	15
	Non-standardized or vague evaluation methods	10

**Table 2 jcm-12-06744-t002:** Search strategies applied for this scoping review.

Combination of Terms Used	Results PUBMED (Total/Relevant Titles)	Results EMBASE (Total/Relevant Titles)
((inlay OR onlay OR veneers OR overlay) AND (“glass ceramics” OR “hybrid ceramics”) AND “survival rate”)	32/10	33/16
((inlay OR onlay OR overlay OR veneers OR partial ceramic restoration) AND “glass ceramic” AND (“polymer-infiltrated ceramic” OR “hybrid ceramic”) AND “survival rate”)	24/9	30/10
((inlay OR onlay OR overlay OR veneers) AND (“glass ceramics” OR “polymer infiltrated ceramics” OR “hybrid ceramics”) AND “survival rate”)	48/19	55/20

**Table 3 jcm-12-06744-t003:** Selected articles and their characteristics.

Author	Year	Types of Restorations (Number of Restorations)	Materials	No. of Patients	Evaluation Criteria	Score Points (Based on the Scoring System)
Spitznagel et al. [25]	2017	Inlays (45); PCR/Onlays (58)	PICN	47	USPHS modified	68/100
Aslan et al. [26]	2019	Veneers (413)	Lithium disilicate	51	USPHS modified	90/100
Coşkun et al. [27]	2019	Inlay (4); Onlay (56)	Lithium disilicate; Hybrid ceramic	14	USPHS modified	60/100
Al-Akhali et al. [28]	2018	Overlay (64)	Lithium disilicate; LS; PICN; PM		Resistance to fracture	40/100
Lu et al. [29]	2017	Onlays (101)	PICN; Feldspathic ceramic	93	USPHS modified	68/100
Taschner et al. [30]	2022	Inlays (70); Onlays (13)	Leucite-reinforced glass ceramics	30	USPHS modified	80/100
Malament et al. [31]	2020	Inlays (246); PCR/Onlays (305)	Lithium disilicate	304	Survival depending on position, age, sex and type of restoration	90/100
Guess et al. [32]	2014	Veneers (66)	Leucite-reinforced glass ceramics	25	USPHS modified	60/100
Xiao et al. [33]	2020	Onlays (120)	Lithium disilicate	120	USPHS modified	75/100
Rinke et al. [34]	2020	Veneers (101)	Leucite-reinforced glass ceramics	31	USPHS modified	70/100
Peumans et al. [35]	2013	Onlays (54); Overlays (8)	Leucite-reinforced glass ceramics	31	USPHS modified	85/100

USPHS: United States Public Health Services; PCR: partial coverage restoration; PICN: polymer infiltrated ceramic network; LS: zirconia-reinforced lithium silicate; PM: polymethylmethacrylate.

**Table 4 jcm-12-06744-t004:** USPHS evaluation criteria.

Criterion	Score	Characteristics
Secondary caries	Alpha	No caries processes
	Bravo	Presence of a carious process
Marginal adaptation	Alpha	No visible defects
	Bravo	Minor defects
	Charlie	Major defects
Joint integrity	Alpha	The joint is not discolored
	Bravo	The joint is superficially discolored
	Charlie	The joint is deeply discolored
Surface condition	Alpha	Smooth and polished surface
	Bravo	Slight visible and palpable roughness
	Charlie	Strong roughness
Color	Alpha	Matching color
	Bravo	Slight mismatch
	Charlie	Strong discordance
Morphology	Alpha	The shape follows the anatomy of the tooth
	Bravo	Slight under- or over-contour
	Charlie	Severely affected anatomy

**Table 5 jcm-12-06744-t005:** Survival results depending on the follow-up duration, the restoration, and the materials taken into account.

Author	Duration of Follow-Up	Restorations (Materials)	Survival in %
Spitznagel et al. [25]	3 years	Inlays	97.4%
		PCR/Onlays (PICN)	95.6%
			Total = 96.5%
Aslan et al. [26]	20 years	Veneers (DL)	98% (5 years)
			95% (10 years)
			91% (15 years)
			87% (20 years)
Coşkun et al. [27]	2 years	Inlays and onlays (DL et CH)	100% (DL)
			100% (CH)
Lu et al. [29]	3 years	Onlays (PICN et CF)	97% (PICN)
			90.7% (CF)
Taschner et al. [30]	14 years	Inlays and onlays	88%
		VL RX	
		VL SV	
Malament et al. [31]	10.9 years	Inlays	93.9%
		PCR/Onlays (DL)	98.3%
Guess et al. [32]	7 years	Veneers (VL)	Total = 98.6%
		FV	100%;
		OV	97.6%
Xiao et al. [33]	2 years	Onlays (DL)	95.83%
Rinke et al. [34]	10 years	Veneers (VL)	91.8%
Peumans et al. [35]	4 years	Inlays and onlays (VL E and VL NE)	97% (E)
			93% (NE)
			Total = 95%

DL: Lithium disilicate; CH: hybrid ceramic; CF: feldspathic ceramic; VL: leucite-reinforced glass ceramic; VL RX: VL glued with a self-adhesive resin; VL SV: VL glued according to a classic protocol; Veneers (VL): Variolink II-low + etching-rinsing; FV: “full veneer preparation”; OV: “modified overlap”; VL E and NE: leucite-reinforced glass ceramic belonging to the “etched” and “non-etched” groups.

**Table 6 jcm-12-06744-t006:** Failures encountered and materials involved.

Author	Absolute Failure/Relative Failure	Materials and Restorations Involved	Type of Failure	Duration after which Failure Occurred
Spitznagel et al. [25]	Absolutes	2 PCR (PICN)	Fractures	23.9 and 28.9 months
		1 inlay (PICN)	Fracture	19.4 months
	Relative	4 PCR (PICN)	Cohesive fractures	11.4; 16.3; 36.9; 38.2 months
Aslan et al. [26]	Absolutes	6 veneers (DL)	Fractures	N/A
		9 veneers (DL)	Detachments	7 between one week and 6 months; 3 to 2 and 5 years
Coşkun et al. [27]	Without failure			
Lu et al. [29]	Absolutes	2 onlays (PICN)	1 Detachment 1 Tooth fracture	12 months
		3 onlays (CF)	1 Fracture of the ceramic 2 Detachments	18 months 24 months
Taschner et al. [30]	Absolutes	4 VL SV	Mass fracture	14 years *
		2 VL RX	Marginal fracture	14 years *
		4 N/A **	N/A	N/A
Malament et al. [31]	Absolutes	3 inlays (DL)	Fracture	2.4 years on average
		3 onlays (DL)	Fracture	
Xiao et al. [32]	Absolutes	5 onlays DL group S	Fractures	2 to 6 months 3 to 1 years
		0 onlays DL group B		
Guess et al. [33]	Absolutes	1 f veneers	Fracture	25 months
	Relative	1 veneers	Detachments	61 months
		12 patients ***	Cohesive fractures	N/A
Rinke et al. [34]	Absolutes ****	10 veneers (VL)	8 Fractures	- 2 : 4 end 10 years - 6 : between 13 and 114 months
			1 Biologic reason	N/A
			1 Changing in the plan of treatment	N/A
	Relative ****	14 veneers (VL)	9 Reglues	N/A
			2 Endodontic treatments	N/A
			2 Caries	97 and 98 months
			2 Minor fractures of the ceramic	N/A
Peumans et al. [35]	Absolutes	2 inlays VL NE	Detachments	6 and 12 months
		1 inlay VL E	Fracture inlay + tooth	48 months

N/A: Not applicable/Not referenced by the author. *: The author indicates that the restoration was clinically unsatisfactory at R7 (14 years), but the last referenced control is at R6 (4 years), so the failure could have occurred over a wide period of 10 years from 4 to 14 years after the restoration was placed. **: Four restorations had to be replaced before R7 (14 years) without the author giving the reason. These failures are likely due to a factor other than the restoration itself (for example, a change in the treatment plan for the tooth concerned by its use as a pillar tooth for a bridge). ***: The author did not specify the number of restorations but specified the number of patients. ****: Failures were considered absolute or relative in all cases, depending on the definition given by the author himself.

## Data Availability

The data presented in this study are available in this manuscript.

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
