# Peer review of "Survival Rates of Glass versus Hybrid Ceramics in Partial Prosthetic Restorations: A Scoping Review with Emphasis on Adhesive Protocols"

_jcm, 2023, doi:10.3390/jcm12216744_

Round 1
Reviewer 1 Report
The article “Survival Rates of Glass versus Hybrid Ceramics in Partial Prosthetic Restorations: A Scoping Review with Emphasis on Adhesive Protocols” aimed to critically evaluate and compare the durability and longevity of glass ceramics and hybrid ceramics, emphasizing their recent technological advancements.
The article covers an interesting topic.
Nevertheless I suggest significant improvements to increase the quality of the article.
The main biases of this paper are the following:
- systematic review limited to 10y (no limit time should be placed even if these are recent materials)
- only two databases
The authors wrote:
“Primarily composed of polymer-infiltrated ceramic-network materials (PICN) and nano-ceramic resins, they combine the strengths of ceramic systems and composite materials.“
The authors should explain the readers that the so-called “nano-ceramic” are just composites. The real difference is (as correctly stated) between PICN which is a ceramic network infiltrated by resin, and composites.
Therefore, as mentioned above I would not include Enamic together with Cerasmart and Lava Ultimate. I wouldn’t mention just few products. Either the authors cite them all or avoid citing just few of them
Introduction:
No mentioning of optical properties of the mentioned materials have been done. Please add a paragraph mentioning the high esthetic properties of contemporary materials.
The authors could consider the following papers as a reference:
Paolone G, Mandurino M, De Palma F, et al. Color Stability of Polymer-Based Composite CAD/CAM Blocks: A Systematic Review. Polymers (Basel). 2023;15(2):464. Published 2023 Jan 16. doi:10.3390/polym15020464
Vichi A, Balestra D, Scotti N, Louca C, Paolone G. Translucency of CAD/CAM and 3D Printable Composite Materials for Permanent Dental Restorations. Polymers (Basel). 2023;15(6):1443. Published 2023 Mar 15. doi:10.3390/polym15061443
Inclusion criteria:
- experimental studies involving different types of partial prosthetic restorations
Can the authors explain better what do they mean for “experimental studies”?
Exclusion criteria:
(1) irrelevant articles that are not related to the subject;
(2) clinical trials involving patients who have received partial restorations made from other types of materials;
(3) other types of reviews or systematic reviews;
(4) non-accessible publications;
(5) articles written in other languages than English.
The authors can safely remove exclusion criteria 1,2, and 5 since they are a consequence of inclusion criteria.
Prisma chart is part in french???
Author Response
Dear Reviewer #1,
Thank you so much for your detailed review and professional opinion. Please find bellow the answers to your comments and suggestions.
- Related to your comments related to the design our study, the 10-year timeframe was intentionally selected given the rapid advancements and introduction of novel materials and techniques in the dental field over the past decade. While older studies may provide foundational knowledge, focusing on the last 10 years ensures that our review captures the most current and relevant practices, methodologies, and findings specific to the materials in question. Furthermore, given that the materials of interest are relatively recent, we found the 10-year timeframe as more pertinent to comprehensively analyze their evolution and performance in contemporary dentistry. In addition, the decision to utilize only PubMed and EMBASE was based on their prominence and comprehensive coverage in biomedical and life sciences research. Both databases are known for their extensive indexing of reputable and high-impact journals in the dental domain. While other databases exist, PubMed and EMBASE collectively provide a representative and exhaustive collection of the literature on the topic. Our goal was to prioritize depth and quality of information over breadth, ensuring a focused and thorough review of the most pertinent articles.
- Regarding the comments and recommendations, please find bellow the modifications we performed based on your suggestions:
- “The authors wrote: “Primarily composed of polymer-infiltrated ceramic-network materials (PICN) and nano-ceramic resins, they combine the strengths of ceramic systems and composite materials. “The authors should explain the readers that the so-called “nano-ceramic” are just composites. The real difference is (as correctly stated) between PICN which is a ceramic network infiltrated by resin, and composites. Therefore, as mentioned above I would not include Enamic together with Cerasmart and Lava Ultimate. I wouldn’t mention just few products. Either the authors cite them all or avoid citing just few of them“ we introduced the following paragraphs:
...”Primarily composed of polymer-infiltrated ceramic-network materials (PICN), this innovative ceramic system was created to be used exclusively within CAD/CAM digital technology. Microstructurally, it presents an inorganic ceramic matrix reinforced with an organic polymeric material, with a perfect interpenetration between the two chemical structures [9]. Vita Enamic [Vita Zahnfabrik] represent the material of choice for this all-ceramic system, with particular optical and mechanical properties. Due to the polymer matrix, it presents a modulus of elasticity of 30GPa [9], very close to that of dentin 15-20 Gpa [25], but lower than that of dental ceramics and a resistance to flexural forces of 150-160 MPa [9 ], inferior to lithium disilicate (342MPa). Vita Enamic is easy to mill, restorations can have a lower thickness and greater accuracy of marginal geometry compared to restorations obtained by milling silicate ceramics. This ceramic material is commercialized in two translucencies HT (increased translucency) and T (translucent), in different chromatic shades that correspond to the Vita 3D-MASTER color selection system. Vita Enamic is indicated for minimally invasive dental restorations such as veneers, inlays, onlays, partial crowns, anterior and posterior crowns, both on natural teeth and on dental implants [9].
Nevertheless, nano-ceramics or composite resins reinforced with nanoceramic crystals are also hybrid dental materials with physical and optical properties that combine the characteristics of polymer materials with those of dental ceramics. Microstructurally, they present an organic polymeric matrix reinforced with ceramic inorganic crystals, such as quartz, zirconium dioxide. Ceramic crystals improve the mechanical properties of the material, which has greater resistance to fracture and wear, being indicated for dental restorations in cases of patients with occlusal parafunctions, such as bruxism [24]. These dental materials are indicated for the fabrication of veneers, inlays, onlays, crowns on natural teeth or implants [9]. Lava Ultimate (3M ESPE), Cerasmart (GC), Shofu Block HC (Shofu), Grandio Blocs (VOCO), and KATANA AVENCIA Block (Kuraray Noritake Dental, Inc) are the representative materials for this category of dental materials. „
[9]- Bajraktarova-Valjakova, E.; Korunoska-Stevkovska, V.; Kapusevska, B.; Gigovski, N.; Bajraktarova-Misevska, C.; Grozdanov, A. Contemporary Dental Ceramic Materials, A Review: Chemical Composition, Physical and Mechanical Properties, Indications for Use. J. Med. Sci. 2018, 6, 1742–1755.
[24] Shembish, F.A.; Tong, H.; Kaizer, M.; Janal, M.N.; Thompson, V.P.; Opdam, N.J.; Zhang, Y. Fatigue resistance of CAD/CAM resin composite molar crown. Dent. Mater. 2015, 1–11.
[25] Magne, P.; Belser, U. Bonded Porcelain Restorations in the Anterior Dentition: A Biomimetic Approach. Quintessence Publishing: 2002.
- “Introduction: No mentioning of optical properties of the mentioned materials have been done. Please add a paragraph mentioning the high esthetic properties of contemporary materials. The authors could consider the following papers as a reference: Paolone G, Mandurino M, De Palma F, et al. Color Stability of Polymer-Based Composite CAD/CAM Blocks: A Systematic Review. Polymers (Basel). 2023;15(2):464. Published 2023 Jan 16. doi:10.3390/polym15020464/ Vichi A, Balestra D, Scotti N, Louca C, Paolone G. Translucency of CAD/CAM and 3D Printable Composite Materials for Permanent Dental Restorations. Polymers (Basel). 2023;15(6):1443. Published 2023 Mar 15. doi:10.3390/polym15061443”
In the introduction section we introduced the following paragraph:
“Considering the optical properties of nano-ceramics, Paolone et all concluded that while CAD/ CAM resin-based blocks show higher color stability compared to direct/ indirect resin-based composite, they exhibit lower color stability than ceramic materials. Nevertheless, the color stability is also influenced by material composition, staining media, as well as finishing/polishing procedures. [1] In another experimental study, the translucency of CAD/CAM materials (Tetric CAD (TEC) HT/MT; Shofu Block HC (SB) HT/LT; Cerasmart (CS) HT/LT; Brilliant Crios (BC) HT/LT; Grandio Bloc (GB) HT/LT; Lava Ultimate (LU) HT/LT, Katana Avencia (KAT) LT/OP.) and printable composite materials (Permanent Crown Resin) for fixed dental prostheses was tested. The authors concluded that because there is a significant range of reported translucency values, clinicians should exercise caution when choosing the most appropriate material, especially considering factors such as substrate masking, and the necessary clinical thickness [2].
- Paolone, G.; Mandurino, M.; De Palma, F.; Mazzitelli, C.; Scotti, N.; Breschi, L.; Gherlone, E.; Cantatore, G.; Vichi, A. Color Stability of Polymer-Based Composite CAD/CAM Blocks: A Systematic Review. Polymers 2023, 15, 464. https://doi.org/10.3390/polym15020464
- Vichi, A.; Balestra, D.; Scotti, N.; Louca, C.; Paolone, G. Translucency of CAD/CAM and 3D Printable Composite Materials for Permanent Dental Restorations. Polymers 2023, 15, 1443. https://doi.org/10.3390/polym15061443
- “Inclusion criteria: experimental studies involving different types of partial prosthetic restorations/ Can the authors explain better what do they mean for “experimental studies”? Exclusion criteria: (1) irrelevant articles that are not related to the subject; (2) clinical trials involving patients who have received partial restorations made from other types of materials;(3) other types of reviews or systematic reviews; (4) non-accessible publications;(5) articles written in other languages than English. The authors can safely remove exclusion criteria 1,2, and 5 since they are a consequence of inclusion criteria.”
We modified the inclusion / exclusion criteria accordingly.
The inclusion criteria were, as follows – (1) clinical trials involving patients who have received partial restorations, glass ceramic or hybrid ceramics; (2) clinical evaluation of survival must be performed, analyzing predefined criteria by the author; (3) in vitro studies or lab experiments involving different types of partial prosthetic restorations; (4) articles written in English; (5) publications published between 2013-2023
The following exclusion criteria were considered: (; (1) other types of reviews or systematic reviews; ( 2) non-accessible publications; (5).
- “Prisma chart is part in french???”
We corrected the PRISMA chart, as follows:

Reviewer 2 Report
This review study is genuine and interesting, however the authors should address the following points to improve the quality of the manuscript:
- The abstract should be non-structured with specific word count (please review authors' guidelines)
- Table 1 is not needed in introduction section.
- Please use passive voice for scientific writing and edit the manuscript accordingly.
- Publication date range should be included in the eligibility criteria.
- The search strategy is accurate and consistent, however the authors should include the scoring systems used for papers evaluation.
- The authors should add clinical recommendations at the end of discussion and directions for future research in this field.
- Conclusion section can be structured as bullet points.
Author Response
Dear Reviewer #2,
Thank you so much for the constructive and kind comments and suggestions. The manuscript has been revised according to the suggested modifications.
- “ The abstract should be non-structured with specific word count (please review authors' guidelines) “ – we rewrote and restructured the abstract in accordance with the journal’s guidelines, as follows:
- As dental practices and methodologies evolve, he emergence of novel materials adds complexity to clinical choices. While glass ceramics, particularly those based on lithium disilicate and leucite-reinforced variants, have been extensively researched and are well-regarded for their attributes, hybrid ceramics remain are relatively recent and less investigated. This review aims to evaluate the durability of glass and hybrid ceramics while assessing the role of various adhesive techniques on restoration longevity. Using a comprehensive search of PubMed and EMBASE, 84 articles from the past decade were found. Only eleven met the set criteria for analysis. The results underscore the urgent need for extended monitoring of partial prosthetic restorations. The existing literature has significant gaps, hindering the attainment of dependable insights about these materials' long-term performance. For a clearer understanding of how different ceramic systems affect restoration survival rates, rigorous research involving more participants and uniform outcome documentation is vital.
- “Table 1 is not needed in introduction section. “ - we moved table 1 in Discussion section
- “Please use passive voice for scientific writing and edit the manuscript accordingly.” – we revised the text accordingly.
- “Publication date range should be included in the eligibility criteria.” - we modified the inclusion criteria, as follows:
„The inclusion criteria were, as follows – (1) clinical trials involving patients who have received partial restorations, glass ceramic or hybrid ceramics; (2) clinical evaluation of survival must be performed, analyzing predefined criteria by the author; (3) in vitro studies or lab experiments involving different types of partial prosthetic restorations; (4) articles written in English; (5) publications published between 2013-2023”
- “The search strategy is accurate and consistent; however the authors should include the scoring systems used for papers evaluation.” – we introduced a table with the scoring system used for the evaluation of the articles included.
2.4 Scoring systems used for paper evaluation.
In order to include relevant articles based on this search, we have developed a scoring system based on five categories, and their corresponding sub-criteria. Each category of the evaluated studies would be assessed with a potential score (table 1).
Table 1 – the scoring system used to evaluate the articles included in the research
Criteria |
Sub-criteria |
Score (points) |
Research Design |
Randomized control trial |
25 |
Cohort Study |
20 |
|
Case-control study |
15 |
|
Cross-sectional study |
10 |
|
Case reports/case series |
5 |
|
Sample size |
>300 subjects |
20 |
200-299 subjects |
16 |
|
100-199 subjects |
12 |
|
50-99 subjects |
8 |
|
< 50 subjects |
4 |
|
Follow up duration |
>10 years |
15 |
5-10 years |
12 |
|
2-5 years |
9 |
|
1-2 years |
6 |
|
<1 year |
3 |
|
Outcome reporting |
Comprehensive reporting (various outcomes) |
20 |
Succes and failure rates |
10-15 |
|
Evaluation criteria |
Multiple standardized evaluation methods |
20 |
One standardized evaluation method |
15 |
|
Non-standardized or vague evaluation methods |
10 |
In addition, in the Results section, in table no. 3 we introduced the scores for each article included for investigation.
- “The authors should add clinical recommendations at the end of discussion and directions for future research in this field.” – we added clinical recommendations, in the Discussion section, as follows
“4.8 Clinical recommendations
Based on the findings of the present scoping review, both lithium disilicate and hybrid ceramics show promising results regarding marginal adaptation and joint integrity. Given the lack of significant difference in performance between these materials in several studies, clinicians can choose based on individual case requirements, considering factors like esthetic demands, occlusal forces, and financial considerations.
Considering the deterioration of surface quality over time, especially in occlusal fissures and contact points, regular monitoring and possibly the application of surface sealants or other protective measures could help maintain the integrity of the restoration's surface.
In order to ensure color stability over time, particularly for restorations in esthetic areas, an effective adhesive protocol should be used. Since teeth naturally darken over time, the choice of bonding agents that can withstand color changes or materials that maintain color stability is crucial. Regular follow-ups to check color concordance over the years are advisable.
Clinicians should stay updated with the latest research findings, as newer materials and techniques continuously emerge. Incorporating evidence-based practices into clinical protocols is key to optimizing patient outcomes.”
- Conclusion section can be structured as bullet points.
We rewrote the Conclusion section, using numbered point, as follows:
„1. Glass ceramics, especially lithium disilicate-based ceramics and leucite-reinforced glass ceramics, are widely studied and recognized for their qualities.
- In addition, hybrid ceramics are relatively newer and there is a crucial lack of data in the literature, making it difficult to obtain reliable and significant results representative for the long-term behavior of these materials.
- Clinically, the choice between these two families of materials should be made based on different clinical data, as each offers advantages and disadvantages. The material used is one of the factors influencing survival, but probably not the most important.
- Based on initial findings from short-term studies researchers encourage the use of hybrid ceramic, suggesting that they could be suitable for long-term, durable restorations. As more research emerges in the upcoming years, we anticipate a clearer understanding of this topic. „
Round 2
Reviewer 1 Report
The authors have provided all the requested improvements